# Security in V2I Communications: A Systematic Literature Review

**DOI:** 10.3390/s22239123

**Published:** 2022-11-24

**Authors:** Pablo Marcillo, Diego Tamayo-Urgilés, Ángel Leonardo Valdivieso Caraguay, Myriam Hernández-Álvarez

**Affiliations:** Departamento de Informática y Ciencias de la Computación, Escuela Politécnica Nacional, Ladrón de Guevara E11-25 y Andalucía, Edificio de Sistemas, Quito 170525, Ecuador

**Keywords:** VANET, V2I, security, privacy, authentication, confidentiality, integrity

## Abstract

Recently, the number of vehicles equipped with wireless connections has increased considerably. The impact of that growth in areas such as telecommunications, infotainment, and automatic driving is enormous. More and more drivers want to be part of a vehicular network, despite the implications or risks that, for instance, the openness of wireless communications, its dynamic topology, and its considerable size may bring. Undoubtedly, this trend is because of the benefits the vehicular network can offer. Generally, a vehicular network has two modes of communication (V2I and V2V). The advantage of V2I over V2V is roadside units’ high computational and transmission power, which assures the functioning of early warning and driving guidance services. This paper aims to discover the principal vulnerabilities and challenges in V2I communications, the tools and methods to mitigate those vulnerabilities, the evaluation metrics to measure the effectiveness of those tools and methods, and based on those metrics, the methods or tools that provide the best results. Researchers have identified the non-resistance to attacks, the regular updating and exposure of keys, and the high dependence on certification authorities as main vulnerabilities. Thus, the authors found schemes resistant to attacks, authentication schemes, privacy protection models, and intrusion detection and prevention systems. Of the solutions for providing security analyzed in this review, the authors determined that most of them use metrics such as computational cost and communication overhead to measure their performance. Additionally, they determined that the solutions that use emerging technologies such as fog/edge/cloud computing present better results than the rest. Finally, they established that the principal challenge in V2I communication is to protect and dispose of a safe and reliable communication channel to avoid adversaries taking control of the medium.

## 1. Introduction

According to Statista [1], by 2021, the number of connected vehicles worldwide will reach 237 million units, and by 2025 that number will be 400 million. The impact of those numbers in the telecommunication area is enormous. One of the implications is related to security in communications. Because of the opening of wireless communications, the dynamic topology and the big size of the network, and the use of the same credentials for registration, attackers may be able to listen, forge, manipulate, or destroy information exchanged between vehicles and roadside units affecting the proper operation and performance of the network [2,3,4].

Generally, there are two modes of communication in a vehicular network: Vehicle to Infrastructure (V2I) and Vehicle to Vehicle (V2V). This last one has the advantage of having roadside units (RSU) with high computational and transmission power to exchange information at high speed with vehicles. For instance, it assures the correct operation of driving guidance and early warning services. In addition to the advantages of V2I, benefits such as avoiding traffic accidents or traffic jams or accessing services on the Internet could justify why many authors have focused more on V2I communication than V2V [2,3]. In that way, it is also essential to cover some aspects of security in this type of communication.

This literature review aims to report the most current and relevant information on V2I communications security. In that way, the authors posed five research questions. The first one is related to the vulnerabilities in this type of communication and their respective countermeasures. The second concerns the methods, technologies, tools, or mechanisms used to provide security solutions to mitigate these vulnerabilities. The third and fourth ones are about the available evaluation metrics to measure the effectiveness of the solutions and thus find out which offers the best results. Finally, the last one concerns the challenges to be faced by the proposals and their possible solutions.

One of the contributions of the present review is the information that the authors provided compared to previous studies related to the same topic. The authors noted that almost all reviews focus on all communication in Vehicular Ad-hoc Network (VANET) without emphasizing V2I communication. VANET is a large overall system model comprising four approaches comprehending driver and vehicle, traffic flow, communications, and applications. The components related to data communication are V2V and V2I. The whole system is vast and complex. Therefore, this research aims to provide a systematic literature review centered on one of the components of VANET: V2I communication, to review it in more detail.

The rest of this article is organized as follows: Section 2 presents the definitions of specific attacks, evaluation metrics, and methods. Meanwhile, Section 3 presents the methodology used to elaborate on this review. Then, Section 4 shows the information used to answer the research questions, followed by Section 5, which introduces the answers to those questions and discusses the most relevant security aspects of V2I. Finally, the conclusions of this work are presented in Section 6.

## 2. Concepts and Definitions

Section 2.1 and Section 2.2 present the lists of the security requirements and attacks that occur in V2I communications. Section 2.3 presents a list of evaluation metrics used by the authors, and Section 2.4 the main methods used to provide solutions.

### 2.1. Security Requirements

Confidentiality: It guarantees that only authorized nodes can access and reveal sensitive information.Integrity: It ensures that the information sent by the sender is the same as that received by the receiver.Authentication: It guarantees that the node that wants to access or use network resources is who it claims to be.Availability: It ensures that the access to network resources for authorized nodes is continuous and without interruptions.Non-repudiation: It guarantees that neither the receiver nor the sender can deny having processed certain information.

### 2.2. Attacks

MitM: It occurs when an adversary secretly enters the communication of two devices to make them believe that they are communicating directly and thus exchange its public key between the devices.Replay: It occurs when an adversary listens to the communication, intercepts it, and later fraudulently resend the obtained messages.Modification/Tampering: It occurs when an adversary alters the message transmitted between two nodes fraudulently.DoS: It occurs when an adversary floods the communication system with no genuine requests getting the service down.Repudiation: It occurs when a system or application does not track nor log the user’s actions properly, permitting manipulation or forging new actions.Session Key Disclosure: It occurs when an adversary can obtain values from memory devices (OBU or TPD) and messages from insecure communication channels. Thus, the adversary can calculate the session key using the values and messages.Impersonation: It occurs when an adversary can take someone’s identity to gain advantages or cause damage to other nodes.Sybil: It occurs when an adversary forges node identities to obtain privileges and thus causes chaos in normal conditions.Forgery: It occurs when an adversary forges a valid certificate and signs a message successfully.Bogus: It occurs when an adversary generates a fake node in a network and informs it about false traffic conditions in a particular location.Eavesdropping: It occurs when an adversary listens to the communication channel extracting information that can be useful for node tracking activity.Plaintext: It occurs when an adversary, who has access to the ciphertext and its plaintext, tries to guess the secret key or develops an algorithm for decrypting messages.Key Leakage: It occurs when an adversary, who has access to the signer, can learn some sensitive information (e.g., computation-time, memory contents, and others).Chosen Message: It occurs when an adversary, who can obtain the ciphertext of plaintext messages from the signer, tries to reveal the secret encryption key.Ciphertext: It occurs when an adversary, who has access to a set of ciphertexts, tries to guess the plaintexts or even the key.Beacon Transmission Denial: It occurs when an adversary suspends itself its beacon transmission for an indefinite time to avoid detection.

### 2.3. Evaluation Metrics

Computational Cost: It refers to the time required to apply certain operations to a message before sending it over the network.Communication Overhead: It refers to the length of information transmitted by a successful message transference.Transmission Delay: It refers to the time a packet takes to get to the destination from the source.Propagation Delay: It refers to the distance between the sender and receiver divided by the light speed.Packet Delivery Ratio: It refers to the ratio of packets successfully delivered to their destinations.Packet Loss Ratio: It refers to the ratio between the number of lost packets and the total number of sent packets.Accuracy: It refers to the general ratio of vehicles correctly detected.Trust Value: It refers to the general cooperativeness of a user.Data Receiving Rate: It refers to the rate of data successfully received.Storage Cost: It refers to the memory size required to store the parameters in the different devices.Roaming Latency: It refers to the time required to transfer the node control between gateways.Cyphertext Length: It refers to the length of messages after performing encryption operations.Energy Consumption: It refers to the energy consumed during the routing process.Throughput: It refers to the rate of messages successfully transmitted in one second over a communication channel.Attack Detection Ratio: It refers to the ratio between the number of attacks detected and the total number of attacks.Average Delay: It refers to the expected time a beacon message of a node remains in a queue before being sent to the infrastructure.False Accept Ratio: It refers to the ratio between the correct number of planned trajectories and the total number of trajectories of a node.

### 2.4. Methods

Elliptic Curve Cryptography (ECC): It is a public key encryption technique that generates cryptographic keys using the elliptic curve theory.Public Key Cryptography (PKC): It is a scheme that performs encryption and decryption using public and private keys. The public key is published, and the private one is kept secret. It is known as asymmetric key cryptography.Symmetric Key Cryptography (SKC): It is a cryptography scheme that uses the same key for encryption and decryption.Public Key Infrastructure (PKI): It is a scheme in which the public key is associated with a certificate provided by a certificate authority instead of choosing one generated randomly.Identity-Based Public Key Cryptography (IBPKC): It is a scheme that uses a representation of identity as the public key to avoid using public ones associated with a certificate. Instead of a certificate authority, there is a key generation center to generate the private keys based on the public ones.Certificateless-Based Cryptography (CBC): It is a scheme that distributes the private keys of the key generation center into several entities. In this scheme, the user and the key generation center calculate the private key, but only the user can obtain the result.

## 3. Materials and Methods

The authors used Kitchenham’s guide [5,6] for performing systematic reviews to elaborate on this work. Thus, they considered the following phases and activities: Planning the Review-Research Questions, Conducting the Review-Search Strategy, Study Quality Assessment, and Data Extraction, and Reporting the Review-Results.

### 3.1. Planning the Review

The five research questions for this review are presented as follows.

RQ01. What are the principal vulnerabilities in V2I communications?RQ02. What are the methods or tools to mitigate those vulnerabilities?RQ03: What evaluation metrics are available to measure the effectiveness of those methods or tools?RQ04: What methods, technologies, or tools provide the best results based on those evaluation metrics?RQ05: What are the principal challenges for mitigating vulnerabilities in VANETs?

### 3.2. Conducting the Review

#### 3.2.1. Search Strategy

The authors used the following bibliographic databases: IEEExplore, ACM Digital Library, and Semantic Scholar. From the research questions, they extracted the following keywords: VANET, V2I, privacy, authentication, confidentiality, and integrity. They developed the search strings using the keywords, the meaning of acronyms VANET and V2I, and the operators AND and OR. Table 1 presents the search strings, date filters, and the search results.

#### 3.2.2. Study Selection

They established some inclusion and exclusion criteria to accomplish the study selection process.

Inclusion Criteria–IC01. Studies that are peer-reviewed research papers.–IC02. Studies published in the last five years.–IC03. Studies published in journals and conferences.Exclusion Criteria–EC01. Studies that are literature reviews, chapters in books, analysis papers, comparative papers, position papers, theses, technical reports, workshop reports, or lecture notes.–EC02. Studies published in preprint platforms.–EC03. The full text of the study is not available.

#### 3.2.3. Quality Assessment

The authors defined one assessment question and established two possible answers for each one. The questions are presented as follows. The answer “no” is rated with 0 and “yes” with 1.

AQ01. Is the study targeted at V2I communications?

They established that the primary study is accepted only if the score is equals to 1; otherwise, it will be rejected. Table A1 presents the quality instrument and its results. As can be seen in Figure 1, 430 articles were found after performing the search. Of them, 48 duplicates were removed, for a total of 382. Then, 335 were removed because they did not meet the inclusion and exclusion criteria, for a total of 47. After that, 11 articles obtained on the snowballing technique were added, for a total of 58. Finally, 11 articles were removed because they did not fulfill the quality criterion. Thus, the selected articles reached 47. Table A2 presents the selected primary studies.

#### 3.2.4. Data Extraction

The authors designed three data collection forms to obtain information from the selected primary studies. The following is a description of each of them. Firstly, Table 2 includes the primary study ID and the attacks to which the proposal is resistant. Whereas Table 3 contains the primary study ID, the category to which it belongs, and the methods, technologies, simulators, and other tools on which the solutions are based. Finally, Table 4 contains the primary study ID and the evaluation metrics used by the proposals. The design of those tables was based on addressing the research questions. Thus, they used Table 2 to answer RQ01, Table 3 to answer RQ02, and Table 4 to answer both RQ03 and RQ04. The generated data are presented in Section 3 and interpreted in Section 4.

### 3.3. Reporting the Review

Of 47 primary studies, 21 are from conferences, and 26 are from journals. The two years when more studies have been published are 2018 and 2019. Figure 2 shows the number of studies and types published by year.

## 4. Results

The authors determined that around seven of every ten solutions use a simulation as an experimental method and around one of every ten solutions as an experiment. Furthermore, they found that one of every two solutions uses network simulators, and one of every two uses both network and traffic simulators. Figure 3 presents the experimental methods and the use of simulators.

### 4.1. Data Collections Forms

The three data collection forms (Table 2, Table 3 and Table 4) described in the Data Extraction are presented as follows. Table 2 presents the attacks to which the proposals are resistant. Table 3 presents the methods, technologies, and tools used to provide solutions. Table 4 presents the evaluation metrics used to measure the effectiveness of the proposals.

### 4.2. Review of Reviews

Intending to contribute more to this topic, the authors performed a simplified review of reviews. Similarly, they used the same search strategy and partly the study selection presented in this section. Below, the authors present two data collection forms (Table 5 and Table 6) with all the information extracted from the reviews.

## 5. Discussion

The answers to the five research questions are presented as follows.

RQ01: What are the principal vulnerabilities in V2I communications?

Since wireless communications are easy to intercept, the principal vulnerability in this type of communication is the susceptibility to attacks. Thus, adversaries can compromise RSUs/vehicles and send false information to drivers putting their lives at risk. They can also send unnecessary alerts to distract them and control the communication links. Once it is done, the adversaries can easily modify session messages. Considering that fact, researchers have focused on proposing solutions that offer any attack resistance. From the results, the authors identified the attacks to which the solutions are resistant. They found the following attacks: MitM, Replay, Tampering, DoS, Repudiation, Session Key Disclosure, Impersonation, Sybil, Forgery, Eavesdropping, and Plaintext. The authors commonly offer solutions against Replay, Impersonation, MitM, Tampering, and Sybil attacks.

RQ02: What methods, technologies, or tools can mitigate those vulnerabilities?

The proposals were grouped using the following categories. The Network Communication Security category for routing protocols, communication schemes, messages exchange security, and privacy protection; the Malicious Node Detection category for intrusion detection systems, trust management schemes, and intrusion prevention systems; and the Authentication Scheme category. According to Figure 4, eight of every ten proposals are about Authentication Schemes, one is about Network Communication Security, and less than one is about Malicious Node Detection.

Regarding technologies, The authors observed that there are several solutions based on PKC/ECC with Blockchain. This technology is gaining attentions in various study fields. This interest must be due to its key features such as decentralization, anonymity, and immutability [46,47]. Regarding simulators, authors have used both network and traffic ones. Figure 5 presents the use of simulators in studies based on the frequency of occurrence. According to it, they use OMNeT++, NS-3, and NS-2 to a greater extent and Veins, SSGA, and MOVE to a lesser extent, and the only traffic simulator is SUMO.

RQ03: What evaluation metrics are available to measure the effectiveness of those methods or tools?

Researchers have used the following evaluation metrics to measure the performance of their solutions. Metrics such as computational cost/time/overhead, communication cost/overhead, storage cost/overhead, transmission delay, propagation delay, packet delivery ratio, packet drop ratio, accuracy, trust value, data receiving rate, roaming latency, cyphertext length, energy consumption, and false success rate. The more common metrics in order of occurrence are computational cost, communication overhead, transmission delay, and packet delivery rate. Because of the use of emerging technologies to solve certain obstacles and limitations, more researchers are focusing on evaluation metrics such as computational cost and communication overhead to measure the effectiveness of their solutions.

RQ04: What methods, technologies, or tools provide the best results based on those evaluation metrics?

From the results, the authors could establish that the solutions that offer better results are the ones in which the use of emerging technologies to overcome certain limits and obstacles are present. Thus, considering metrics such as computational cost and overhead, the solutions based on Fog/Edge/Cloud computing present better results than the others. The following comparative analysis (Table 7) reinforces this assumption.

From the main methods (Figure 6), there is a slight trend of using Elliptic Curve Cryptography (ECC) instead of traditional cryptography (PKC); however, the evaluation metrics present good results for both cases. In this case, it is necessary for further research to determine the best method based on the evaluation metrics. Apart from the methods, researchers have also used network and traffic simulators, map tools, security tools, programming languages, platforms, and libraries. Figure 7 presents the distribution of the tools used in the proposals. The most used map tool is Open Street. About security tools, the most common are MIRACL and Avispa. The most used programming languages are C and Python, and among libraries, OpenSSL.

RQ05: What are the principal challenges for mitigating vulnerabilities in VANETs?

Since vehicles with limited computing resources must interact with communication infrastructure at high speed, the great challenge in vehicular networks is to dispose of a safe and reliable communication channel and suitable device performance. When a vehicle enters the coverage of a new roadside unit, the computational overhead can lower the quality of communications and driving safety. Not all attacks in vehicular networks are protected with security mechanisms such as cryptography techniques, digital signatures, or message verification technique, and there are others as the bogus attack that requires a solution. Counting with a secure channel to transmit authentication information is still a paradigm considering that some security schemes must be applied to resource-limited and time-critical devices.

## 6. Conclusions and Future Work

The capacity of modern vehicles to connect to an external infrastructure makes them vulnerable to cyber-attacks. Counting with a secure channel to transmit authentication information is still a paradigm considering that some security schemes must be applied to resource-limited and time-critical devices. V2I communications offer more advantages and benefits to users than V2V communications. Hence, the reasons for studying the state-of-the-art of security in V2I communications.

In the present review, the authors found that the principal attacks to which solutions are resistant are multiple and varied. The attackers could intrude on a network to intercept and manipulate the messages using MitM, Replay, Repudiation, Eavesdropping, or Tampering; shut down a machine making it inaccessible with DoS; pretend to be someone else to access information through impersonation and Sybil attack; tricks a web browser into executing unwanted actions using Forgery; obtain the key with plaintext attacks where the attacker knows the plaintext and its corresponding encrypted ciphertext or with Exploitation of the session control mechanisms with Session Key disclosure; among others fraudulent techniques.

Diverse methods and tools are developed to mitigate these vulnerabilities. They grouped them into solutions for authentication/trust management/network communication schemes for privacy preserving, IDS and IPS models to alert and act over a security incident, and routing protocol management to protect the devices. To evaluate the effectiveness of the methods and tools used to mitigate the vulnerabilities that measure computational cost through communication overhead, transmission delay, and data delivery reliability. The authors observed that researchers must address their future work toward using emerging technologies to reduce computational overhead and save computational costs. They observed a slight trend in using ECC instead of traditional cryptography. However, it is too soon to establish if ECC will become the dominant choice in cryptography in a few years. What is certain is that the use of emerging technologies such as Fog/Edge/Cloud computing, Cloudlets, Blockchain, Software-Defined Networking, and Network Functions Virtualization has suffered a rapid expansion. In fact, the inclusion of emerging technologies in proposals has contributed to reducing the computational overhead and saving the computational costs.

After conducting a simplified review of reviews, the authors observed that the number of analyzed articles in almost all reviews is insufficient, and the lack of a search strategy is surprising. The reviews vaguely mentioned evaluation metrics and emerging technologies as possible solutions to overcome certain limitations. Concerning the methods used to build solutions, the other authors mentioned at least the most common ones. Finally, the list of threats/attacks proposed in the reviews is relatively small compared to the list in the present review.

On the other hand, the authors identified coincidences in the presentation of information on fundamentals, security requirements, threats/attacks, solutions, and challenges. However, the present review stands out because it presents, for instance, a comparative analysis of emerging technologies in relation to some performance metrics, some graphics related to the percentage of occurrence in solutions of the methods, tools, and simulators used by researchers to build solutions against vulnerabilities, and also one representing the percentage of occurrence about the types of solutions presented in the present review. Not to mention the valuable information the authors obtained from a review of review articles.

## Figures and Tables

**Figure 1 sensors-22-09123-f001:**
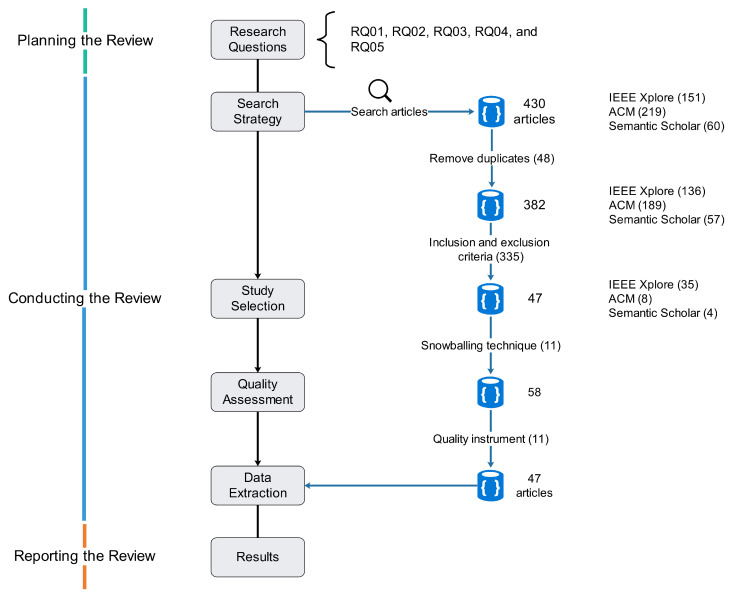
The review process.

**Figure 2 sensors-22-09123-f002:**
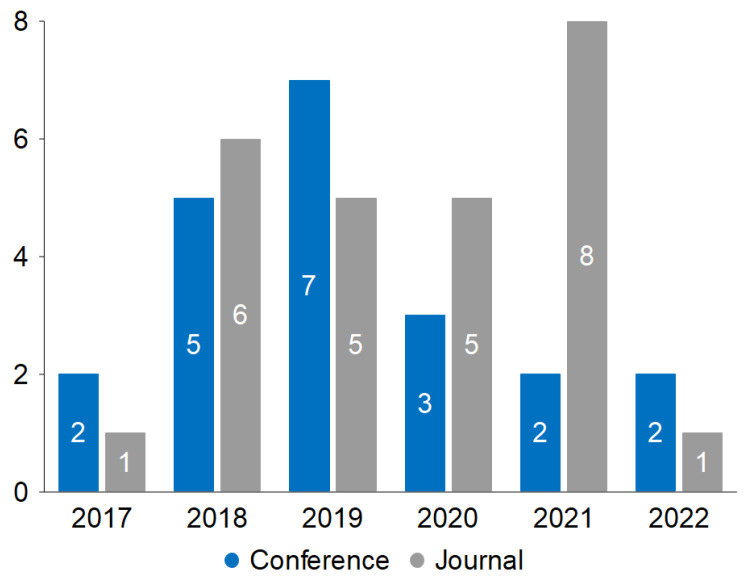
Primary studies published by year and type of publication.

**Figure 3 sensors-22-09123-f003:**
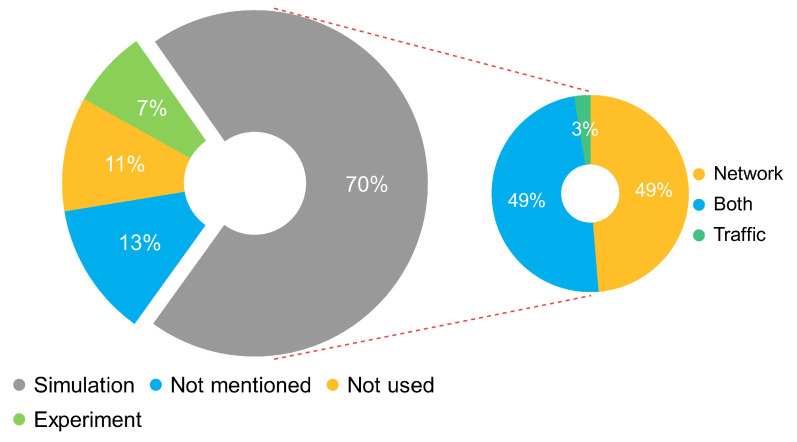
Experimental methods and use of simulators based on the frequency of occurrence.

**Figure 4 sensors-22-09123-f004:**
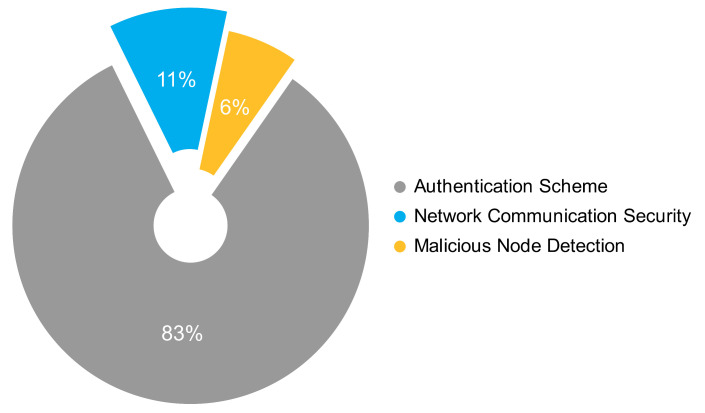
Types of solutions based on the frequency of occurrence.

**Figure 5 sensors-22-09123-f005:**
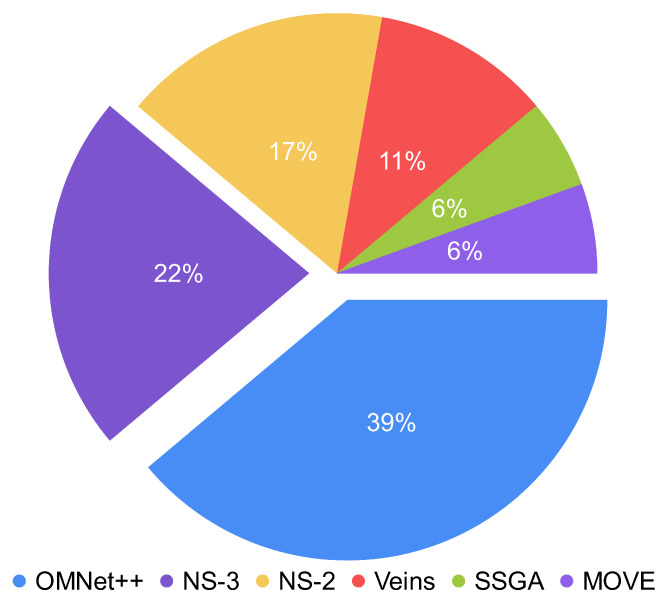
Use of network simulators based on the frequency of occurrence.

**Figure 6 sensors-22-09123-f006:**
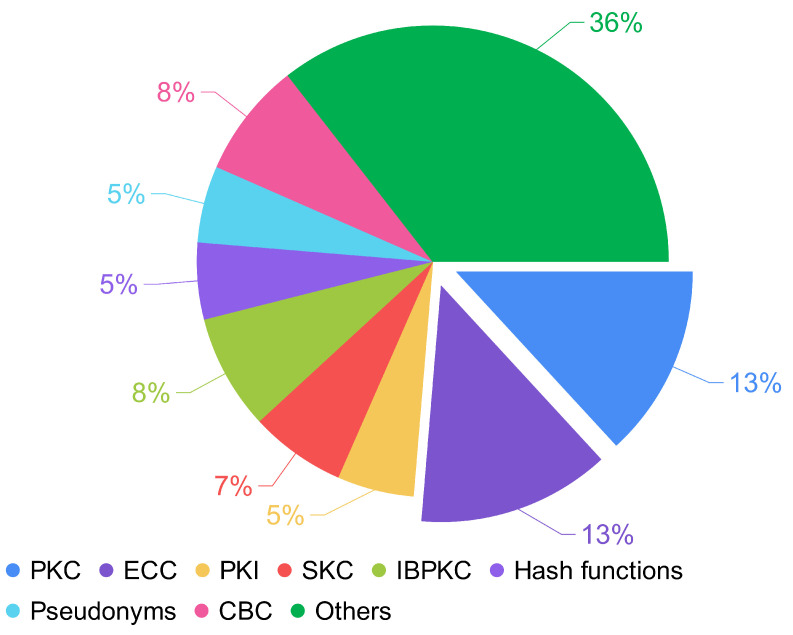
Use of methods based on the frequency of occurrence.

**Figure 7 sensors-22-09123-f007:**
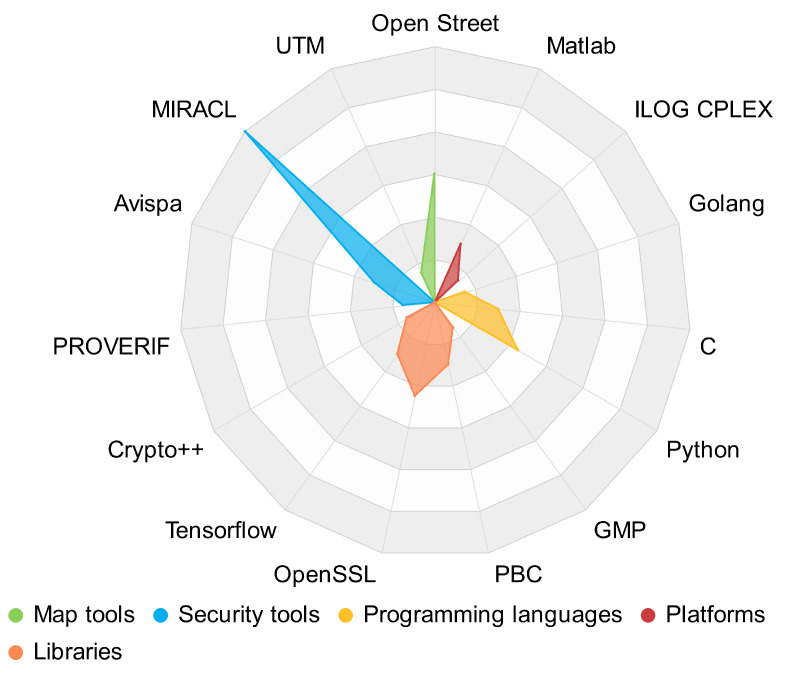
Use of tools based on the frequency of occurrence.

**Table 1 sensors-22-09123-t001:** Search Results.

Database Search Engine	ID	Command Search	Date Filters	Search Date	Total
Scopus	SS01	((“All Metadata”:”vehicular ad hoc network” OR “All Metadata”:vanet) AND (“All Metadata”:”vehicle to infrastructure” OR “All Metadata”:v2i) AND “All Metadata”:security AND (“All Metadata”:privacy OR “All Metadata”:authentication OR “All Metadata”:confidentiality OR “All Metadata”:integrity))	2017–2022	22 June 2022	55
	SS02	((“All Metadata”:”vehicle to infrastructure” OR “All Metadata”:v2i) AND “All Metadata”:security AND (“All Metadata”:privacy OR “All Metadata”:authentication OR “All Metadata":confidentiality OR “All Metadata”:integrity))	2017–2022	22 June 2022	96
					151
ACM	SS03	[[All: “vehicular ad hoc networks”] OR [All: vanet]] AND [[All: “vehicle to infrastructure”] OR [All: v2i]] AND [All: security] AND [[All: privacy] OR [All: authentication] OR [All: confidentiality] OR [All: integrity]]	[Publication Date: Past 5 years]	22 June 2022	67
	SS04	[[All: “vehicle to infrastructure”] OR [All: v2i]] AND [All: security] AND [[All: privacy] OR [All: authentication] OR [All: confidentiality] OR [All: integrity]]	[Publication Date: Past 5 years]	22 June 2022	152
					219
Semantic Scholar	SS05	vanet vehicular ad-hoc network v2i vehicle to infrastructure privacy authentication confidentiality integrity	Last 5 years	22 June 2022	40
	SS06	v2i vehicle to infrastructure privacy authentication confidentiality integrity	Last 5 years	22 June 2022	20
					60

**Table 2 sensors-22-09123-t002:** Protection against attacks.

ID	MiTM	Replay	Modification	Tampering	DoS	Session Key Disclosure	Impersonation	Sybil	Forgery	Bogus	Eavesdropping	Plaintext	Key Leakage	Chosen Message	Ciphertext	Beacon Trans. Denial
PS01	√	√					√									
PS02	√	√	√				√									
PS03	√	√	√				√									
PS04							√	√								
PS05				√												
PS06		√		√			√									
PS07													√			
PS08		√														
PS09 *																
PS10	√	√														
PS11		√							√							
PS12													√			
PS13		√	√				√									
PS14	√	√														
PS15		√	√				√			√						
PS16 *																
PS17	√	√					√									
PS18				√												
PS19			√				√	√								
PS20	√	√					√						√			
PS21																√
PS22				√												
PS23	√	√	√				√									
PS24	√	√	√				√									
PS25											√					
PS26			√				√	√			√					
PS27		√														
PS28		√		√												
PS29	√	√	√				√									
PS30	√															
PS31														√		
PS32		√														
PS33	√	√	√			√	√	√								
PS34											√					
PS35														√	√	
PS36		√														
PS37										√						
PS38	√	√	√													
PS39	√	√	√	√	√		√		√							
PS40							√						√			
PS41	√	√	√			√	√					√				
PS42												√			√	
PS43										√						
PS44			√		√											
PS45		√	√		√		√	√								
PS46 *																
PS47										√						

* Not defined. √ Defined.

**Table 3 sensors-22-09123-t003:** Methods and tools.

ID	Category	Methods	Based on Technology	Simulators	Other Tools
PS01	Authentication	Elliptic Curve Cryptography (ECC)	Blockchain	✗	Bouncy Castle [7] and Scyther Tool [8]
PS02	Authentication	Certificateless-Based Cryptography (CBC)	Blockchain	✗	MIRACL [9]
PS03	Authentication	Cuckoo filters	✗	✗	Crypto++ [10]
PS04	Authentication	Public Key Cryptography (PKC) and Symmetric Key Cryptography	✗	NS-2 [11]	✗
PS05	Network Security	Identity-Based PKC	Blockchain	OMNeT++ [12] and SUMO [13]	✗
PS06	Authentication	Public Key Infrastructure (PKI) and ECC	Blockchain	✗	✗
PS07	Authentication	PKC	Blockchain	✗	Crypto++
PS08	Authentication	PKC and ECC	Blockchain	✗	Go environment [14]
PS09	Authentication	PKC and Certificateless-Based Cryptography	✗	✗	Java Pairing Based Cryptography (JPBC) library [15]
PS10	Authentication	PKC and Trust Degree Estimation	✗	✗	Matlab [16]
PS11	Authentication	Hash functions and XOR operations	✗	✗	✗
PS12	Authentication	ECC and Private Key Insulation	✗	✗	TEPLA [17]
PS13	Authentication	Certificateless-Based Cryptography and ECC	✗	✗	MIRACL
PS14	Authentication	PKC	Blockchain	✗	GMP [18] and PBC [19] libraries
PS15	Authentication	Certificateless-Based Cryptography	✗	NS-3 [20] and SUMO	✗
PS16	Authentication	PKC	✗	OMNeT++ and SUMO	✗
PS17	Authentication	Protocols suite	✗	OMNeT++ and SUMO	Tamarin [21], TEPLA [17], and Python Cryptography Tool (pycrypto) [22]
PS18	Intrusion Prevention System	✗	Edge Computing, Software, Defined Networking (SDN) and Network Functions Virtualization (NFV)	✗	Stop Sign Gap Assist (SSGA) [23]
PS19	Authentication	Identity-Based PKC and Hash Message Authentication Code (HMAC)	✗	NS-3 [20], SUMO, and MObility Model Generator for Vehicular Networks (MOVE) [24]	✗
PS20	Authentication	✗	Blockchain	NS-3, Automated Validation of Security Protocols and Applications (AVISPA) [25]	MIRACL and PBC
PS21	Intrusion Detection System	Auto Correlation Function (ACF)	✗	OMNeT++ and SUMO	✗
PS22	Network Security	Port Hopping Technique	✗	✗	✗
PS23	Authentication	PKC and Identity-Based PKC	✗	NS-2	✗
PS24	Authentication	Certificateless-Based Cryptography	✗	✗	MIRACL
PS25	Authentication	Cooperative Group Beaconing Strategy	✗	✗	✗
PS26	Authentication	Protocol Based on Vehicles Trajectory	✗	✗	Open Street Map [26]
PS27	Authentication	Identity-Based PKC and Pseudonyms	✗	✗	MIRACL and PBC
PS28	Authentication	PKC	✗	✗	AVISPA and OpenSSL [27]
PS29	Authentication	Lattice-Based Cryptosystem	✗	✗	✗
PS30	Authentication	Roaming Protocol	✗	OMNeT++	✗
PS31	Authentication	ECC, Identity-based PKC, and Pseudonyms	✗	✗	MIRACL
PS32	Authentication	Symmetric Encryption and Group Signature	✗	✗	✗
PS33	Communication Protocol	ECC and One-Way Hash Function	✗	✗	Testbed [28]
PS34	Authentication	BGN Homomorphic Encryption and Moore Curve Technique	✗	✗	✗
PS35	Authentication	PKI and Identity-Based Cryptosystem (IBC)	✗	✗	✗
PS36	Authentication	Certificateless Aggregate Signature and Pseudonyms	✗	✗	MIRACL
PS37	Authentication	Symmetric Key Encryption	Fog Computing	✗	✗
PS38	Authentication	Authenticated Key Agreement (AKA), ECC and Hash Functions	✗	OMNeT++, SUMO, and Veins [29]	MIRACL and Crypto++
PS39	Authentication	Identity-Based Cryptography	✗	✗	✗
PS40	Authentication	Reputation-Based Validation	Blockchain	✗	Socket.io [30], Node.js [31], and Google Maps [32]
PS41	Authentication	Symmetric and Asymmetric Cryptography, and ECC	✗	✗	AVISPA and Proverif [33]
PS42	Messages exchange security	Permutation-Only Encryption	Group Formation Criteria	✗	✗
PS43	Messages exchange security	✗	Cloudlets	✗	AWS IoT [34], AWS Greengrass [35], and Boto [36]
PS44	Authentication	PKI and Message Accessing Code (MAC) Encryption	✗	✗	✗
PS45	Authentication	Authentication Tokens and Schnorr Signature	✗	NS-3 and SUMO	✗
PS46	Privacy Protection	Trusted Third Party (TTP) and Circle Based Dummy Generation (CBDG)	✗	OMNeT++, SUMO, and Veins	✗
PS47	Intrusion Detection System (IDS)	Position Verification Technique	✗	NS-2	✗

✗ Not defined.

**Table 4 sensors-22-09123-t004:** Evaluation Metrics.

ID	Computational Cost	Communication Overhead	Transmission Delay	Propagation Delay	Packet Delivery Ratio	Packet Loss Ratio	Accuracy	Trust Value	Data Receiving Rate	Storage Cost	Storage Overhead	Roaming Latency	Cyphertext Length	Energy Consumption	Throughput	Attack Detection Rate	Average Delay	False Accept Rate	Others
PS01	√	√	√	√															
PS02	√	√																	
PS03	√																		
PS04	√																		
PS05	√				√	√													
PS06	√	√	√																
PS07	√										√								
PS08	√	√																	
PS09	√																		
PS10	√				√		√												
PS11	√																		
PS12	√																		
PS13	√	√																	
PS14	√							√											
PS15	√	√	√			√													
PS16		√	√		√														
PS17	√																		
PS18									√										
PS19	√	√																	
PS20	√														√				
PS21																√			
PS22																√			
PS23	√	√																	
PS24	√	√									√								
PS25																	√		
PS26																		√	
PS27	√																		
PS28	√																		
PS29	√	√								√									
PS30		√										√							
PS31	√	√																	
PS32	√																		
PS33	√	√								√				√					
PS34	√																		
PS35	√	√								√			√						
PS36	√	√																	
PS37	√	√																	
PS38	√	√																	
PS39	√	√																	
PS40																		√	
PS41 *																			
PS42																			√
PS43																			√
PS44 *																			
PS45		√																	
PS46																			√
PS47					√														

* Not defined. √ Defined.

**Table 5 sensors-22-09123-t005:** Review articles.

ID	Authors	Title	Aim	Search Strategy	Number of Studies	Year
RV01	Islam et al. [37]	A Comprehensive Survey on Attacks and Security Protocols for VANETs	It informs about fundamentals, application areas, security services, and possible attacks in vehicular networks.	Not defined	11	2021
RV02	Hamdi et al. [38]	A review on various security attacks in vehicular ad hoc networks	It discusses security concerns, security services, and advances in authentication. It also describes attacks and threats.	Not defined	13	2021
RV03	Sheikh et al. [39]	A Survey of Security Services, Attacks, and Applications for Vehicular Ad Hoc Networks (VANETs)	It presents an overview of VANET that includes architecture, communication methods, standards, and characteristics. It also presents security services, security threats and attacks, simulation tools, and challenges.	Not defined	>65	2019
RV04	Singh et al. [40]	Advanced Security Attacks on Vehicular AD HOC Network (VANET)	It presents security requirements, challenges, attacks, and privacy issues in VANET.	Not defined	Not defined	2019
RV05	Azam et al. [41]	An outline of the security challenges in VANET	It informs about security attacks and solutions.	Not defined	6	2020
RV06	Kohli et al. [42]	Future Trends of Security and Privacy in Next Generation VANET	It addresses security and privacy issues in next-generation VANET. It also presents solutions for those issues.	Not defined	11	2020
RV07	Mitsakis et al. [43]	Recent Developments on Security and Privacy of V2V & V2I Communications: A Literature Review	It presents solutions to both the attacks and challenges in a VANET.	Not defined	15	2020
RV08	Mihai et al. [44]	Security Aspects of Communications in VANETs	It presents relevant proposals for privacy, authentication, and integrity in the context of vehicular networks.	Not defined	12	2020
RV09	Goyal et al. [45]	Security Attacks, Requirements, and Authentication Schemes in VANET	It provides a classification of attacks, security requirements, and authentication schemes	Not defined	8	2019
N/A	Our review	Security in V2I Communications: A Systematic Literature Review	It informs the principal vulnerabilities and challenges in V2I communications, the tools and methods to mitigate those vulnerabilities, the evaluation metrics to measure the effectiveness of those tools and methods, and based on those metrics, the methods or tools that provide the best results.	Available	47	2022

**Table 6 sensors-22-09123-t006:** Comparative analysis among reviews.

ID	Protection against Attacks	Methods and Tools	Based on Technology	Simulators	Evaluation Metrics
RV01	DoS	PKI			
	Tampering	ECC			
	Sybil	IBPKC			
	Replay				
	Impersonation				
RV02	DoS		Blockchain		
	Tampering				
	Impersonation				
	Sybil				
	Replay				
	Eavesdropping				
	MitM				
RV03	DoS	PKC	Cloud computing	SUMO	Computational cost
	Tampering	SKC		OMNET++	Communication overhead
	MitM	IBPKC		NS-2	
	Eavesdropping	PKI		NS-3	
	Impersonation	Hash functions		Veins	
	Replay	ECC			
	Sybil				
RV04	Tampering				
	Impersonation				
	Sybil				
	MitM				
	DoS				
	Eavesdropping				
	Replay				
	Bogus				
RV05	DoS	PKI	Blockchain		
	Tampering	Hash functions	Fog computing		
	Eavesdropping				
	Sybil				
	Replay				
RV06	DoS	PKI			
	Bogus				
	Impersonation				
	Eavesdropping				
RV07	DoS	PKI			
	Sybil	PKC			
	MitM	SKC			
	Tampering				
	Impersonation				
	Replay				
RV08	MitM	PKI	Blockchain		
	Impersonation	IBPKC			
	Tampering				
	Eavesdropping				
RV09	Tampering	PKI			Computational cost
	Eavesdropping	ECC			Communication overhead
		SKC			Average delay
		IBPKC			
Our review	MitM	ECC	Blockchain	SUMO	Computational cost
	Replay	CBC	Edge computing	OMNet++	Communication overhead
	Modification	PKI	SDN	NS-3	Transmission delay
	Tampering	PKC	NFV	NS-2	Propagation delay
	DoS	SKC	Fog computing	Veins	Packet Delivery Ratio
	Session Key Disclosure	IBPKC	Group Formation Criteria	SSGA	Packet Loss Ratio
	Impersonation	Hash functions	Cloud computing	MOVE	Accuracy
	Sybil	Pseudonyms			Trust Value
	Forgery				Data Receiving Rate
	Bogus				Storage Cost
	Eavesdropping				Storage Overhead
	Plaintext				Roaming Latency
	Key Leakage				Cyphertext Length
	Chosen Message				Energy Consumption
	Ciphertext				Throughput
	Beacon Transmission Denial				Attack Detection Rate
					Average Delay
					False Accept Rate

**Table 7 sensors-22-09123-t007:** Comparative analysis of emerging technologies in relation to some performance metrics.

	Blockchain	Fog	Edge	Cloud	Cloudlets
		Computing			
Latency	Low	Medium	Low	High	Low
Scalibility	Low	High	High	Medium	Low
Energy Consumption	High	Medium	Low	High	Medium
Interoperability	Low	High	Low	High	Low

## Data Availability

Not applicable.

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
