# Peer review of "Security in V2I Communications: A Systematic Literature Review"

_sensors, 2022, doi:10.3390/s22239123_

Round 1

Reviewer 1 Report

The authors present a literature review related to Security in V2I Communication.

There are various shortcames:

1. Limited/No declaration of "why" the paper is important in the sector.

 Where are similar paper that addresses this topic? 

2.  No description or references to the considered attacks

3. The answers to the five research questions (RQ) are generic and not clear

4.  RQ04 answer is partial

5. RQ5 is addressed only in qualitative way (not quantitative)

6. The first line of the paper refer to a prediction from 2015 related to 2020 (The authors should use more recent prediction e.g. statista.com Connected cars worldwide - statistics & facts)

 7.  PS09 PS16 PS31 PS46 are not classified in Table 4, why?

The same for Table 6: PS40-43 and PS46

8. Evaluation metrics are not described (Trust Value is not clear at all)

9. Since a lot of papers use more than one simulator, the Figure 5 is not clear.

10. False success rate = False positive rate?

11. In RQ04 and Figure 6 tool, and programming languages are merged together (no description of methods is present)

Reviewer 2 Report

1.       What is the reason to do this review? How this review makes progress in the field? What is new aspect of this review in relation to the field?

2.       Additional presentations: 6G Driven Fast Computational Networking Framework for Healthcare Applications, A blockchain based lightweight peer-to-peer energy trading framework for secured high throughput micro-transactions, Hybrid Technique for Cyber-Physical Security in Cloud-Based Smart Industries, Energy Efficient Consensus Approach of Blockchain for IoT Networks with Edge Computing.

3.       What is your research hypothesis? What do you want to find in your research? Make them clearly stated in the beginning and solved in conclusion.

4.       Compare technologies in advances and disadvantages. Show advances of each technology.

Reviewer 3 Report

The authors presented a systematic literature review related to security in V2I communication.  The followings are the suggestions to the authors:

1. Table 2 and Table 3 seems not necessary. 

2. The complete manuscript should be explained as third person. In the abstract: "The authors have identified the .... " but also written as "Thus, we found schemes resistant to attacks .... ". This should not be accepted.

3. Table 1 does not provide any important knowledge. 

4. In some rows of some table there is no data but only the reference of the paper. Those fields should be removed. 

5. In the whole paper, there are only a few contributions in the discussion section, in the other parts of the paper, the contribution is not significant at all. 

6. There are several review papers related to the authors' selected topic. Thus, it requires a comparative analysis between those and should present the contribution of the submitted manuscript as well as its novel contributions.

However, the topic is an updated topic and the paper selection is also very good. There are lots of opportunities to focus and possibly summarize adequately. Please search for related review articles in similar fields and try to provide more deep information about the topic as well as reviewed papers.

Round 2

Reviewer 1 Report

The authors reviewed the paper.

Some notes:

1. The first reference was not modified with the new one (Statista report)

2. Figure 5 reports twice NS3 simulator.

3.The following sentence can be improved "This literature review aims to inform the most current and relevant about security in communications"

Reviewer 2 Report

paper is revised

Reviewer 3 Report

Thank you for the response.

Description of attack, evaluation metric, and method can not be considered as contributions. Actually  it requires some recently published review works related to your field and compared to those what are the novel contributions of yours.  Most preferably in a table format where there will be a list of review papers and their speciality in different columns. For example in the left column write the specifications and then compare those with your proposed system. 
